# STATE ALIGNMENT-BASED IMITATION LEARNING

**Fangchen Liu**    **Zhan Ling**    **Tongzhou Mu**    **Hao Su**
University of California San Diego
La Jolla, CA 92093, USA
`{fliu,z6ling,t3mu,haosu}@eng.ucsd.edu`

## ABSTRACT

Consider an imitation learning problem that the imitator and the expert have different dynamics models. Most of the current imitation learning methods fail because they focus on imitating actions. We propose a novel state alignment based imitation learning method to train the imitator to follow the state sequences in expert demonstrations as much as possible. The state alignment comes from both local and global perspectives and we combine them into a reinforcement learning framework by a regularized policy update objective. We show the superiority of our method on standard imitation learning settings and imitation learning settings where the expert and imitator have different dynamics models.

## 1 INTRODUCTION

Learning from demonstrations (imitation learning, abbr. as IL) is a basic strategy to train agents for solving complicated tasks. Imitation learning methods can be generally divided into two categories: behavior cloning (BC) and inverse reinforcement learning (IRL). Behavior cloning (Ross et al., 2011b) formulates a supervised learning problem to learn a policy that maps states to actions using demonstration trajectories. Inverse reinforcement learning (Russell, 1998; Ng et al., 2000) tries to find a proper reward function that can induce the given demonstration trajectories. GAIL (Ho & Ermon, 2016) and its variants (Fu et al.; Qureshi et al., 2018; Xiao et al., 2019) are the recently proposed IRL-based methods, which uses a GAN-based reward to align the distribution of state-action pairs between the expert and the imitator.

Although state-of-the-art BC and IRL methods have demonstrated compelling performance in standard imitation learning settings, e.g. control tasks (Ho & Ermon, 2016; Fu et al.; Qureshi et al., 2018; Xiao et al., 2019) and video games (Aytar et al., 2018b), these approaches are developed based on a strong assumption: the expert and the imitator share **the same** dynamics model; specifically, they have the same action space, and any feasible state-action pair leads to the same next state in probability for both agents. The assumption brings severe limitation in practical scenarios: Imagine that a robot with a low speed limit navigates through a maze by imitating another robot which moves fast, then, it is impossible for the slow robot to execute the exact actions as the fast robot. However, the demonstration from the fast robot should still be useful because it shows the path to go through the maze.

We are interested in the imitation learning problem under a relaxed assumption: Given an imitator that shares the same state space with the expert but their dynamics may be different, we train the imitator to follow the state sequence in expert demonstrations as much as possible. This is a more general formulation since it poses fewer requirements on the experts and makes demonstration collection easier. Due to the dynamics mismatch, the imitator becomes more likely to deviate from the demonstrations compared with the traditional imitation learning setting. Therefore, it is very important that the imitator should be able to resume to the demonstration trajectory by itself. Note that neither BC-based methods nor GAIL-based IRL methods have learned to handle dynamics misalignment and deviation correction.

To address the issues, we propose a novel approach with four main features: 1) *State-based*. Compared to the majority of literature in imitation learning, our approach is state-based rather than action-based. Not like BC and IRL that essentially match state-action pairs between the expert and the imitator, we only match states. An inverse model of the imitator dynamics is learned to recover the action; 2) *Deviation Correction*. A state-based $\beta$-VAE (Higgins et al., 2017) is learned as the prior for the next state to visit. Compared with ordinary behavior cloning, this VAE-based next state predictor can advise the imitator to return to the demonstration trajectory when it deviates. The

robustness benefits from VAE's latent stochastic sampling; 3) *Global State Alignment*. While the VAE can help the agent to correct its trajectory to some extent, the agent may still occasionally enter states that are far away from demonstrations, where the VAE has no clue how to correct it. So we have to add a global constraint to align the states in demonstration and imitation. Inspired by GAIL that uses reward to align the distribution of state-action pairs, we also formulate an IRL problem whose maximal cumulative reward is the Wasserstein Distance between states of demonstration and imitation. Note that we choose not to involve state-action pairs as in GAIL(Ho & Ermon, 2016), or state-state pairs as in an observation-based GAIL (Torabi et al., 2018a), because our state-only formulation imposes weaker constraints than the two above options, thus providing more flexibility to handle different agent dynamics; 4) *Regularized Policy Update*. We combine the prior for next state learned from VAE and the Wasserstein distance-based global constraint from IRL in a unified framework, by imposing a Kullback-Leibler divergence based regularizer to the policy update in the Proximal Policy Optimization algorithm.

To empirically justify our ideas, we conduct experiments in two different settings. We first show that our approach can achieve similar or better results on the standard imitation learning setting, which assumes the same dynamics between the expert and the imitator. We then evaluate our approach in the more challenging setting that the dynamics of the expert and the imitator are different. In a number of control tasks, we either change the physics properties of the imitators or cripple them by changing their geometries. Existing approaches either fail or can only achieve very low rewards, but our approach can still exhibit decent performance. Finally, we show that even for imitation across agents of completely different actuators, it is still possible for the state-alignment based method to work. Surprisingly, a point mass and an ant in MuJoCo (Todorov et al., 2012) can imitate each other to navigate in a maze environment.

Our contributions can be summarized as follows:

- Propose to use a state alignment based method in the imitation learning problems where the expert's and the imitator's dynamics are different.
- Propose a local state alignment method based on $\beta$-VAE and a global state alignment method based on Wasserstein distance.
- Combine the local alignment and global alignment components into a reinforcement learning framework by a regularized policy update objective.

## 2 RELATED WORK

Imitation learning is widely used in solving complicated tasks where pure reinforcement learning might suffer from high sample complexity, like robotics control (Le et al., 2017; Ye & Alterovitz, 2017; Pathak et al., 2018), autonomous vehicle (Fu et al.; Pomerleau, 1989), and playing video game (Hester et al., 2018; Pohlen et al., 2018; Aytar et al., 2018a). Behavioral cloning (Bain & Sommut, 1999) is a straight-forward method to learn a policy in a supervised way. However, behavioral cloning suffers from the problem of compounding errors as shown by (Ross & Bagnell, 2010), and this can be somewhat alleviated by interactive learning, such as DAGGER (Ross et al., 2011b). Another important line in imitation learning is inverse reinforcement learning (Russell, 1998; Ng et al., 2000; Abbeel & Ng, 2004; Ziebart et al., 2008; Fu et al.), which finds a cost function under which the expert is uniquely optimal.

Since IRL can be connected to min-max formulations, works like GAIL, SAM (Ho & Ermon, 2016; Blondé & Kalousis, 2018) utilize this to directly recover policies. Its connections with GANs (Goodfellow et al., 2014) also lead to $f$-divergence minimization (Ke et al., 2019; Nowozin et al., 2016) and Wasserstein distance minimization (Xiao et al., 2019). One can also extend the framework from matching state-action pairs to state distribution matching, such as Torabi et al. (2018a); Sun et al. (2019); Schroecker & Isbell (2017). Other works (Aytar et al., 2018b; Liu et al., 2018; Peng et al., 2018) also learn from observation alone, by defining reward on state and using IRL to solve the tasks. Works like (Lee et al., 2019; Lee et al.) also use state-based reward for exploration. Torabi et al. (2018b); Edwards et al. (2018) will recover actions from observations by learning an inverse model or latent actions. However, our work aims to combine the advantage of global state distribution matching and local state transition alignment, which combines the advantage of BC and IRL through a novel framework.

## 3 BACKGROUNDS

**Variational Autoencoders** Kingma & Welling (2013); Rezende et al. (2014) provides a framework to learn both a probabilistic generative model $p_\theta(\mathbf{x}|\mathbf{z})$ as well as an approximated posterior

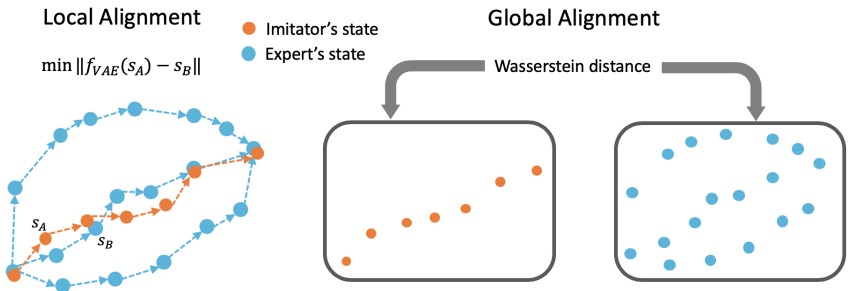

Figure 2: Visualization of state alignment

distribution $q_\phi(\mathbf{z}|\mathbf{x})$. $\beta$-VAE is a variant VAE that introduces an adjustable hyperparameter $\beta$ to the original objective:

$$\mathcal{L}(\theta, \phi; \mathbf{x}, \mathbf{z}, \beta) = \mathbb{E}_{q_\phi(\mathbf{z}|\mathbf{x})}\left[\log p_\theta(\mathbf{x}|\mathbf{z})\right] - \beta D_{KL}\left(q_\phi(\mathbf{z}|\mathbf{x})\|p(\mathbf{z})\right) \quad (1)$$

Larger $\beta$ will penalize the total correlation (Chen et al., 2018) to encourage more disentangled latent representations, while smaller $\beta$ often results in sharper and more precise reconstructions.

**Wasserstein distance**   The Wasserstein distance between two density functions $p(x)$ and $q(x)$ with support on a compact metric space $(M, d)$ has an alternative form due to Kantorovich-Rubenstein duality (Villani, 2008):

$$\mathcal{W}(p, q) = \sup_{\phi \in \mathcal{L}_1} \mathbb{E}_{p(x)}[\phi(x)] - \mathbb{E}_{q(x)}[\phi(x)] \quad (2)$$

Here, $\mathcal{L}_1$ is the set of all 1-Lipschitz functions from $\mathcal{M}$ to $\mathbb{R}$. Compared with the prevalent KL-divergence and its extension, the f-divergence family, Wasserstein distance has a number of advantages theoretically and numerically. Please refer to Arjovsky et al. (2017) and Solomon (2018) for a detailed discussion.

# 4   SAIL: STATE ALIGNMENT BASED IMITATION LEARNING

## 4.1   OVERVIEW

Our imitation learning method is based on state alignment from both local and global perspectives. For local alignment, the goal is to follow the transition of the demonstration as much as possible, and allow the return to the demonstration trajectory whenever the imitation deviates. To achieve both goals, we use a $\beta$-VAE (Higgins et al., 2017) to generate the next state (Figure 2 Left). For global alignment, we set up an objective to minimize the Wasserstein distance between the states in the current trajectory and the demonstrations (Figure 2 Right). There has to be a framework to naturally combine the local alignment and global alignment components. We resort to the reinforcement learning framework by encoding the local alignment as policy prior and encoding the global alignment as reward over states. Using Proximal Policy Optimization (PPO) by Schulman et al. (2017) as the backbone

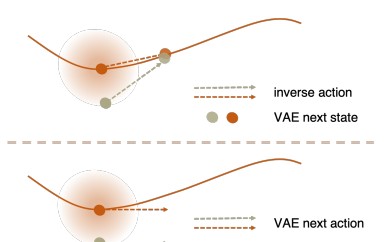

Figure 1: Using VAE as a state predictive model will be more self-correctable because of the stochastic sampling mechanism. But this won't happen when we use VAE to predict actions.

RL solver, we derive a regularized policy update. To maximally exploit the knowledge from demonstrations and reduce interactions with the environment, we adopt a pre-training stage to produce a good initialization based on the same policy prior induced by the local alignment. Our method is summarized in Algorithm 1. In the rest parts of this section, we will introduce all the components of our method in details.

## 4.2   LOCAL ALIGNMENT BY STATE PREDICTIVE VAE

To align the transition of states locally, we need a predictive model to generate the next state which the agent should target at. And then we can train an inverse dynamics model to recover the cor-

---

**Algorithm 1** SAIL: State Alignment based Imitation Learning

---

**Require:** Expert trajectories $\tau_e : [s_1, a_1, s_2, a_2, ...] \sim \pi_e$, initial policy $\pi$, inverse dynamics model $g$, discriminator $\phi$, total episode $T$, memory capacity $S$
1: **if** Imitator and Expert have the same dynamics model **then**
2:     Pre-train $g$ using $\tau_e$ and transitions collected by a random policy
3: **else**
4:     Pre-train $g$ using transitions collected by a random policy
5: **end if**
6: Pre-train VAE using $\tau_e$, and obtain the policy prior         ▷ Pre-train VAE and obtain policy prior
7: Pretrain $\pi$ using policy prior as described in Sec 4.5
8: **while** episode $\leq$ T **do**
9:     **while** $|\tau| \leq S$ **do**                             ▷ $\tau$ is the collected trajectories
10:         Collect trajectory $\{(s, a, s', r, done)\}$ using $\pi$
11:         Update $r$ using (4)
12:         Add $\{(s, a, s', r, done)\}$ to $\tau$
13:     **end while**
14:     Train $\phi$ using $\max_{\phi \in \mathcal{L}_1} E_{s \sim \tau_e}[\phi(s)] - E_{s \sim \tau}[\phi(s)]$     ▷ Calculate Wasserstein Distance
15:     Update inverse dynamics model $g$
16:     Update policy using (5)
17: **end while**

---

responding action, so as to provide a direct supervision for policy. It is worth-noting that, while training an inverse dynamics model is generally challenging, it is not so hard if we only focus on the agent dynamics, especially when the low-dimensional control states are accessible as in many practical scenarios. The problem of how to learn high-quality inverse/forward dynamics models is an active research topic.

Instead of using an ordinary network to memorize the subsequent states, which will suffer from the same issue of compounding errors as behavioral cloning (Ross & Bagnell, 2010; Ross et al., 2011a), we propose to use VAE to generate the next state based on the following two reasons. First, as shown in (Dai et al., 2018), VAE is more robust to outliers and regularize itself to find the support set of a data manifold, so it will generalize better for unseen data. Second, because of the latent stochastic sampling, the local neighborhood of a data point will have almost the same prediction, which is self-correctable when combined with a precise inverse dynamics model as illustrated in Figure 1.

We can also use a VAE to generate action based on the current state. But if the agent deviated from the demonstration trajectory a little bit, this predicted action is not necessarily guide the agent back to the trajectory, as shown in Figure 1. And in Sec 5.3.2, we conduct experiments to compare the state predictive VAE and the action predictive VAE.

Instead of the vanilla VAE, we use $\beta$-VAE to balance the KL penalty and prediction error, with formulation shown in (1). In Sec 5, we discuss the effects of the hyper-parameter $\beta$ in different experiment settings as one of the ablation studies.

### 4.3 GLOBAL ALIGNMENT BY WASSERSTEIN DISTANCE

Due to the difference of dynamics between the expert and the imitator, the VAE-based local alignment cannot fully prevent the imitator from deviating from demonstrations. In such circumstances, we still need to assess whether the imitator is making progress in learning from the demonstrations. We, therefore, seek to control the difference between the state visitation distribution of the demonstration and imitator trajectories, which is a global constraint.

Note that using this global constraint alone will not induce policies that follow from the demonstration. Consider the simple case of learning an imitator from experts of the same dynamics. The expert takes cyclic actions. If the expert runs for 100 cycles with a high velocity and the imitator runs for only 10 cycles with a low velocity within the same time span, their state distribution would still roughly align. That is why existing work such as GAIL aligns state-action occupancy measure. However, as shown later, our state-based distribution matching will be combined with the local alignment component, which will naturally resolve this issue. The advantage of this state-based distribution matching over state-action pair matching as in GAIL or state-next-state pair matching in (Torabi et al., 2018a) is that the constraint becomes loosened.

We use IRL approach to achieve the state distribution matching by introducing a reinforcement learning problem. Our task is to design the reward to train an imitator that matches the state distribution of the expert.

Before introducing the reward design, we first explain the computation of the Wasserstein distance between the expert trajectories $\{\tau_e\}$ and imitator trajectory $\{\tau\}$ using the Kantorovich duality:

$$\mathcal{W}(\tau_e, \tau) = \sup_{\phi \in \mathcal{L}_1} \mathbb{E}_{s \sim \tau_e}[\phi(s)] - \mathbb{E}_{s \sim \tau}[\phi(s)] \tag{3}$$

where $\phi$ is the Kantorovich's potential, and serves as the discriminator in WGAN (Arjovsky et al., 2017). $\phi$ is trained with a gradient penalty term as WGAN-GP introduced in (Gulrajani et al., 2017)

After the rollout of imitator policy is obtained, the potential $\phi$ will be updated by (3). Assume a transition among an imitation policy rollout of length $T$ is $(s_i, s_{i+1})$. To provide a dense signal every timestep, we assign the reward as:

$$r(s_i, s_{i+1}) = \frac{1}{T}[\phi(s_{i+1}) - \mathbb{E}_{s \sim \tau_e}\phi(s)] \tag{4}$$

We now explain the intuition of the above reward. By solving (3), those states of higher probability in demonstration will have a larger $\phi$ value. The reward in (4) will thus encourage the imitator to visit such states.

Maximizing the curriculum reward will be equivalent to

$$J(\pi) = \sum_{t=1}^{T} \mathbb{E}_{s_t, s_{t+1} \sim \pi}[r(s_t, s_{t+1})] = \sum_{t=1}^{T} \frac{\mathbb{E}_{s_{t+1}}[\phi(s_{t+1}) - \mathbb{E}_{s \sim \tau_e}[\phi(s)]]}{T} = -\mathcal{W}(\tau_e, \tau)$$

In other words, the optimal policy of this MDP best matches the state visitation distributions w.r.t Wasserstein distance.

Compared with AIRL (Fu et al.) that also defines rewards on states only, our approach indeed enjoys certain advantages in certain cases. We provide a theoretical justification in the Appendix D.

## 4.4 REGULARIZED PPO POLICY UPDATE OBJECTIVE

As mentioned in the second paragraph of Sec 4.3, the global alignment has to be combined with local alignment. This is achieved by adding a prior to the original clipped PPO objective.

We maximize the following unified objective function:

$$J(\pi_\theta) = L^{CLIP}(\theta) - \lambda D_{KL}\left(\pi_\theta(\cdot|s_t)\,\middle\|\, p_a\right) \tag{5}$$

We will explain the two terms in detail. $L^{CLIP}(\theta)$ denotes the clipped surrogate objective used in the original PPO algorithm:

$$L^{CLIP}(\theta) = \hat{\mathbb{E}}_t\left[\min\left(\frac{\pi_\theta(a|s)}{\pi_{\theta_{old}}(a|s)}\hat{A}_t, \text{clip}\left(\frac{\pi_\theta(a|s)}{\pi_{\theta_{old}}(a|s)}, 1 - \epsilon, 1 + \epsilon\right)\hat{A}_t\right)\right], \tag{6}$$

where $\hat{A}_t$ is an estimator of the advantage function at timestep $t$. The advantage function is calculated based on a reward function described in Sec 4.3.

The $D_{KL}$ term in (5) serves as a regularizer to keep the policy close to a learned policy prior $p_a$. This policy prior $p_a$ is derived from the state predictive VAE and an inverse dynamics model. Assume the $\beta$-VAE is $f(s_t) = s_{t+1}$ and the inverse dynamics model is $g_{inv}(s_t, s_{t+1}) = a$. To solve the case when the agents have different dynamics, we learn a state prediction network and use a learned inverse dynamics to decode the action. We define the action prior as

$$p_a(a_t|s_t) \propto \exp\left(-\left\|\frac{g_{inv}(s_t, f(s_t)) - a_t}{\sigma}\right\|^2\right) \tag{7}$$

where the RHS is a pre-defined policy prior, a Gaussian distribution centered at $g_{inv}(s_t, f(s_t))$. $\sigma$ controls how strong the action prior is when regularizing the policy update, which is a hyperparameter. Note that the inverse model can be further adjusted during interactions.

$L^{CLIP}$ is computed through the advantage $\hat{A}_t$ and reflects the global alignment. The policy prior is obtained from the inverse model and local $\beta$-VAE, which makes the $D_{KL}$ serve as a local alignment constraint. Furthermore, our method can be regard as a combination of BC and IRL because our KL-divergence based action prior encodes the BC policy and we update the policy leveraging reward.

We would note that our state-alignment method augments state distribution matching by taking relationships of two consecutive states into account with robustness concern.

## 4.5 PRE-TRAINING

We pretrain the state predictive VAE and the inverse dynamics model, and then obtain the policy prior in (7), which is a Gaussian distribution. For pre-training, we want to initialize PPO's Gaussian policy $\pi$ by this prior $p_a$, by minimizing the KL-divergence between them. Practically, we use direct supervision from $g_{inv}(s_t, f(s_t))$ and $\sigma$ in (7) to directly train both the mean and variance of the policy network, which is more efficient during the pre-training stage. During the online interaction, the update rule of PPO's policy is by optimizing (5), and the variance will be further adjusted for all the dimensions of the action space.

## 5 EXPERIMENTS

We conduct two different kinds of experiments to show the superiority of our method. In Sec 5.1, we compare our method with behavior cloning (Bain & Sommut, 1999), GAIL (Ho & Ermon, 2016), and AIRL (Fu et al.) in control setting where the expert and the imitator have different dynamics model, e.g., both of them are ant robots but the imitator has shorter legs. In Sec 5.1, we further evaluate in the traditional imitation learning setting. Finally, in Sec 5.3, we conduct ablation study to show the contribution of the components.

## 5.1 IMITATION LEARNING ACROSS AGENTS OF DIFFERENT ACTION DYNAMICS

### 5.1.1 ACTORS OF MODIFIED PHYSICS AND GEOMETRY PROPERTIES

We create environments using MuJoCo (Todorov et al., 2012) by changing some properties of experts, such as density and geometry of the body. We choose 2 environments, Ant and Swimmer, and augment them to 6 different environments: Heavy/Light/Disabled Ant/Swimmer. The Heavy/Light agents have modified density, and the disabled agents have modified head/tail/leg lengths. The demonstrations are collected from the standard Ant-v2 and Swimmer-v2. More descriptions of the environments and the demonstration collection process can be founded in the Appendix.

We then evaluate our method on them.

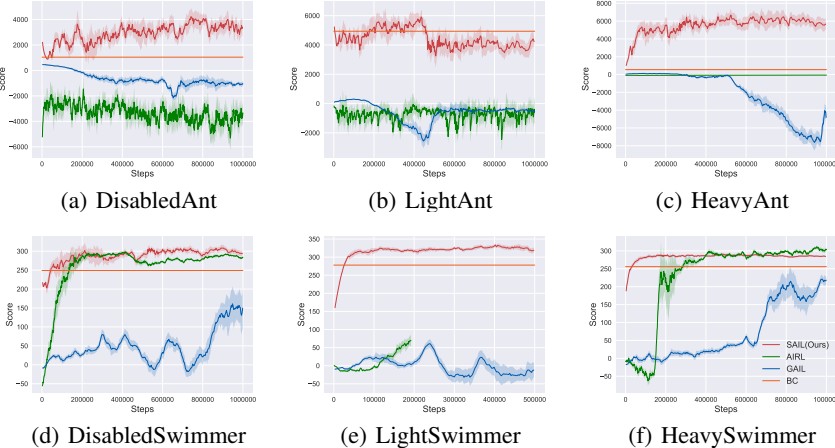

(a) DisabledAnt      (b) LightAnt      (c) HeavyAnt

(d) DisabledSwimmer      (e) LightSwimmer      (f) HeavySwimmer

Figure 3: Comparison with BC, GAIL and AIRL when dynamics are different from experts.

Figure 3 demonstrates the superiority of our methods over all the baselines. Our approach is the most stable in all the 6 environments and shows the leading performance in each of them. GAIL seems to be the most sensitive to dynamics difference. AIRL, which is designed to solve imitation learning for actors of different dynamics, can perform on par with our method in two swimmer-based environments (DisabledSwimmer and HeavySwimmer) that have relatively lower dimensional action space (2D for swimmer versus 8D for ants).

Interestingly, the stability and performance of vanilla behavior cloning are quite reasonable in 4 of the environments, although it failed to move about in the DisabledAnt and HeavyAnt environments. For these two tasks, the agent will reach dangerous states by cloning actions, yet our method will not approach these states by using state-based imitation. In the other four games, BC agents do not die but just move less efficiently, so they have a sub-optimal yet still reasonable score. [1]

### 5.1.2 ACTORS OF HETEROGENEOUS ACTION DYNAMICS

We consider an extremely challenging setting that the imitator and demonstrator are functionally different. One typical example of expert/imitator pair in practice would be a human and a humanoid robot. We consider a much simplified version but with similar nature – a Point and an Ant in MuJoCo. In this task, even if the state space cannot be exactly matched, there are still some shared dimensions across the state space of the imitator and the actor, e.g., the location of the center of mass, and the demonstration should still teach the imitator in these dimensions.

We use the same setting as many hierarchical RL papers, such as HIRO and Near-Optimal RL (Nachum et al., 2018a;b). The agent need to reach a goal position in a maze, which is represented by (x,y) coordinates. We also know that the first two dimensions of states are the position of the agent. The prior knowledge includes: (1) the goal space (or the common space that need to be matched) (2) the projection from the state space to the goal space (select the first two dimensions of the states).

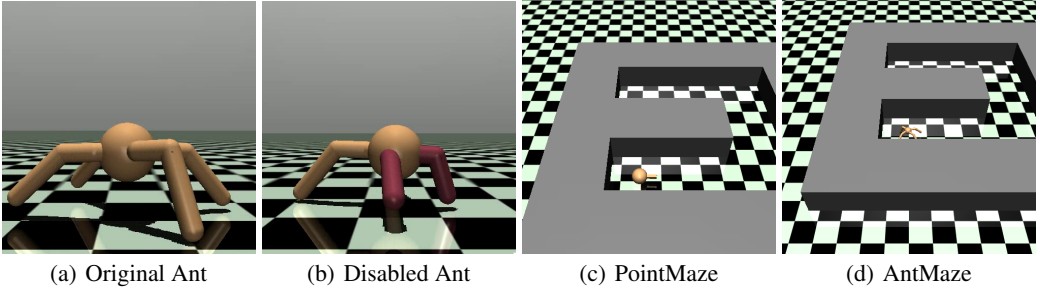

(a) Original Ant          (b) Disabled Ant          (c) PointMaze          (d) AntMaze

Figure 4: Imitation Learning of Actors with Heterogeneous Action Dynamics.

The first task is that the Ant should reach the other side of the maze from several successful demonstrations of a Point robot. As shown in Figure 4(c) and Figure 4(d), the maze structure for the ant and point mass is exactly the same.

To solve this problem, we first pre-train an VAE on the demonstrations, and use this VAE to propose the next "subgoal" for the Ant. This VAE is trained on the goal space (i.e. the first two dimensions) of the Point robot's trajectory. Then we train an inverse model for Ant, which will generate an action based on the Ant's current state (high dimensional) and goal predicted by VAE (2 dimensional).

Our performance is shown in Figure 5(c). After 1M training steps, the agent has success rate of 0.8 to reach the other side of the maze.

### 5.2 ACTORS OF THE SAME DYNAMICS (STANDARD IMITATION LEARNING)

We also evaluate our algorithm on 6 non-trivial control tasks in MuJoCo: Swimmer, Hopper, Walker, Ant, HalfCheetach, and Humanoid. We first collect demonstration trajectories with Soft Actor-Critic, which can learn policies that achieve high scores in most of these environments[2]. For com-

---

[1]For LightSwimmer 3(e), AIRL meets MuJoCo numerical exception for several trials.

[2]We collect near-optimal demonstration on Swimmer using TRPO due to the limited performance of SAC.

parison, we evaluate our method against 3 baselines: behavior cloning, GAIL, and AIRL[3]. Also, to create even stronger baselines for the cumulative reward and imitator run-time sample complexity, we initialize GAIL with behavior cloning, which would obtain higher scores in Swimmer and Walker. Lastly, to evaluate how much each algorithm depends on the amount of demonstrations, we sampled demonstration trajectories of ten and fifty episodes.

Table 1 depicts representative results in Hopper and HalfCheetah[4]. The advantage of our methods over BC should be attributed to the inherent data augmentation by VAE. On Hopper-v2, we are significantly better with 10 demos but are just on par if the demos are increased to 50. On HalfCheetah-v2, the demo cheetah runs almost perfectly ( 12294 scores); in other words, the demo provides limited instruction when the imitator is even slightly off the demo states, thus the robustness from VAE becomes critical.

Table 1: Performance on Hopper-v2 and HalfCheetah-v2

| | Hopper-v2 | | HalfCheetah-v2 | |
|---|---|---|---|---|
| # Demo | 10 | 50 | 10 | 50 |
| Expert | $3566 \pm 1.24$ | | $12294.22 \pm 273.59$ | |
| BC | $1318.76 \pm 804.36$ | $3525.87 \pm 160.74$ | $971.42 \pm 249.62$ | $4813.20 \pm 1949.26$ |
| GAIL | $3372.66 \pm 130.75$ | $3363.97 \pm 262.77$ | $474.42 \pm 389.30$ | $-175.83 \pm 26.76$ |
| BC-GAIL | $3132.11 \pm 520.65$ | $3130.82 \pm 554.54$ | $578.85 \pm 934.34$ | $1597.51 \pm 1173.93$ |
| AIRL | $3.07 \pm 0.02$ | $3.31 \pm 0.02$ | $-146.46 \pm 23.57$ | $755.46 \pm 10.92$ |
| Our init | $3412.58 \pm 450.97$ | $3601.16 \pm 300.14$ | $1064.44 \pm 227.32$ | $7102.29 \pm 910.54$ |
| Our final | $\mathbf{3539.56 \pm 130.36}$ | $\mathbf{3614.19 \pm 150.74}$ | $\mathbf{1616.34 \pm 180.76}$ | $\mathbf{8817.32 \pm 860.55}$ |

## 5.3 ABLATION STUDY

### 5.3.1 COEFFICIENT $\beta$ IN $\beta$-VAE

$\beta$-VAE introduces an additional parameter to the original VAE. It controls the variance of the randomly sampled latent variable sampling, which subsequently affects the reconstruction quality and robustness. Theoretically, a smaller $\beta$ leads to better state prediction quality, with the cost of losing the deviation correction ability (Dai et al., 2018).

To empirically show the role of beta and check the sensitivity of our algorithm with respect to beta, we evaluate VAE in settings of both the imitator has the same dynamics and has different dynamics. We select HalfCheetah-v2 and HeavyAnt as an example. For HalfCheetah-v2, we pretrain the inverse dynamics and VAE using given demonstrations so that the initial performance will tell the quality of the VAE's prediction. For DisabledAnt, we pretrain the dynamics with random trials, which results in forward/inverse dynamics estimation of less accuracy. In this case, we examine both its initialized performance and final performance. The results are shown in Table 2. We find out that for $\beta$ in $[0.01, 0.1]$, the performance is better. Specifically, when the imitator is different from the expert, a smaller $\beta$ will result in poor performance as it overfits the demonstration data.

We also compare our method with an ordinary MLP trained by MSE loss. We find out that VAE outperforms MLP in all settings. Note that the MLP-based approach is very similar to the state-based behavior cloning work of (Torabi et al., 2018b).

### 5.3.2 ACTION PREDICTIVE $\beta$-VAE

In Figure 1, we mentioned that a VAE to predict the next action is less favorable. To justify the claim, we compare a VAE-based BC with a vanilla BC that both predict actions, as shown in Table 3. Experiments show that VAE-BC is even outperformed by a vanilla BC, especially when $\beta$ is larger than 0.001. Compared with the last line in Table 2, we can conclude that VAE is more useful when predicting state, which consolidates that the advantage really comes from our state-based approach but not only the robustness of VAE.

### 5.3.3 EFFECT OF WASSERSTEIN DISTANCE AND KL REGULARIZATION

In our policy update process, we use Wasserstein distance with KL regularization to update the policy. To analyze their effects on the performance, we use HalfCheetah-v2 and Humanoid-v2 with

---

[3]AIRL and EAIRL(Qureshi et al., 2018) have similar performance, and we only compare to AIRL.
[4]Results for other environments can be founded in the Appendix.

Table 2: Analyze the role of VAE coefficient. The "None" item means replacing VAE with an ordinary network with linear layers.

| $\beta$ | Environments | | | |
|---|---|---|---|---|
| | HalfCheetah-50 | HalfCheetah-20 | HeavyAnt-Initial | HeavyAnt-Final |
| 0.2 | 2007.86 | 1289.21 | 258.91 | 282.13 |
| 0.15 | 2653.04 | 1149.65 | 1149.65 | 1502.68 |
| 0.1 | **7102.29** | 1797.44 | **1219.34** | **5208.45** |
| 0.05 | 5933.28 | **2215.71** | 987.72 | 4850.62 |
| 0.01 | 5893.17 | 1982.62 | 740.54 | 1921.26 |
| 0.005 | 4415.04 | 1369.57 | 320.54 | 399.31 |
| None | 4759.69 | 1123.79 | 359.15 | -62.13 |

Table 3: Compare behavior cloning to variational behavior cloning

| $\beta$ | Environments | |
|---|---|---|
| | HalfCheetah-50 | Hopper-50 |
| 0.1 | $230.52 \pm 13.26$ | $203.87 \pm 14.39$ |
| 0.01 | $1320.04 \pm 15.43$ | $438.10 \pm 20.43$ |
| 0.001 | $3306.91 \pm 12.51$ | $3303.72 \pm 10.46$ |
| None | $\mathbf{4813.20 \pm 1949.26}$ | $\mathbf{3525.87 \pm 6.74}$ |

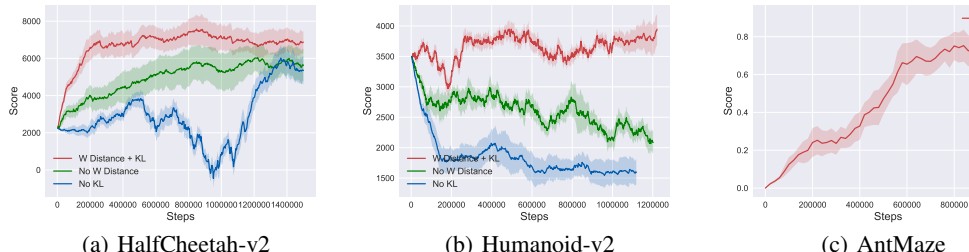

(a) HalfCheetah-v2      (b) Humanoid-v2      (c) AntMaze

Figure 5: (a), (b) show the effects of Wasserstein distance and KL regularization on HalfCheetah-v2 and Humanoid-v2 given 20 demonstration trajectories. And (c) presents the result on Antmaze.

20 expert trajectories. For each environment, they use the same pretrained inverse model and VAE, thus they have the same behavior after pretraining.

As shown in Figure 5(a)(b), Wasserstein distance combined with KL regularization performs the best. Wasserstein objective is used in our inverse RL based mechanism that would significantly penalize the exploration when the agent deviates from the demonstration far away. However, using this objective alone lacks constraints over consecutive states, thus performing the worst. The KL objective adds constraints over consecutive states using a VAE prior; however, VAE is unable to extrapolate to states when the imitator deviates far from the demo (green line gradually fails as in Fig 5 (b)), but this is the scenario when the Wasserstein distance would not favor, thus the reward from the Wasserstein distance will push the imitator back to the demonstration states.

## 6    CONCLUSION

We proposed SAIL, a flexible and practical imitation learning algorithms that use state alignment from both local and global perspective. We demonstrate the superiority of our method using MuJoCo environments, especially when the action dynamics are different from the demonstrations.

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

# A    LEARNING ACROSS DIFFERENT ENVIRONMENTS

**PointMaze & AntMaze** As shown in Figure 4, a point mass or an ant is put in a $24 \times 24$ U-maze. The task is to make the agent reach the other side of U-maze with the demonstration from the point mass. The ant is trained to reach a random goal in the maze from a random location, and should reach the other side of the maze. The state space of ant is 30-dim, which contains the positions and velocities.

**HeavyAnt** Two times of original Ant's density. Two times of original gear of the armature.

**LightAnt** One tenth of original Ant's density.

**DisabledAnt** Two front legs are 3 quarters of original Ant's legs.

**HeavySwimmer** 2.5 times of original Swimmer's density.

**LightSwimmer** One twentieth of original Swimmer's density.

**DisabledSwimmer** Make the last joint 1.2 times longer and the first joint 0.7 times of the original length

The exact results of these environments are listed in Table 4, 5. All the statistics are calculated from 20 trails.

Table 4: Performance on modifeid Swimmer

|  | DisabledSwimmer | LightSwimmer | HeavySwimmer |
|---|---|---|---|
| BC | $249.09 \pm 1.53$ | $277.99 \pm 3.41$ | $255.95 \pm 2.5$ |
| GAIL | $228.46 \pm 2.02$ | $-4.11 \pm 0.51$ | $254.91 \pm 1.35$ |
| AIRL | $283.42 \pm 3.69$ | $67.58 \pm 25.09$ | $\mathbf{301.27 \pm 5.21}$ |
| SAIL(Ours) | $\mathbf{287.71 \pm 2.31}$ | $\mathbf{342.61 \pm 6.14}$ | $286.4 \pm 3.2$ |

Table 5: Performance on modified Ant

|  | DisabledAnt | HeavyAnt | LightAnt |
|---|---|---|---|
| BC | $1042.45 \pm 75.13$ | $550.6 \pm 77.62$ | $\mathbf{4936.59 \pm 53.42}$ |
| GAIL | $-1033.54 \pm 254.36$ | $-1089.34 \pm 174.13$ | $-971.74 \pm 123.14$ |
| AIRL | $-3252.69 \pm 153.47$ | $-62.02 \pm 5.33$ | $-626.44 \pm 104.31$ |
| SAIL(Ours) | $\mathbf{3305.71 \pm 67.21}$ | $\mathbf{5608.47 \pm 57.67}$ | $4335.46 \pm 82.34$ |

# B    IMITATION BENCHMARK EXPERIMENTS SETTINGS AND RESULTS

We use six MuJoCo (Todorov et al., 2012) control tasks. The name and version of the environments are listed in Table 6, which also list the state and action dimension of the tasks with expert performance and reward threshold to indicate the minimum score to solve the task. All the experts are trained by using SAC (Haarnoja et al., 2018) except Swimmer-v2 where TRPO (Schulman et al., 2015) get higher performance.

Table 6: Performance on benchmark control tasks

| Environment | State Dim | Action Dim | Reward threshold | Expert Performance |
|---|---|---|---|---|
| Swimmer-v2 | 8 | 2 | 360 | 332 |
| Hopper-v2 | 11 | 3 | 3800 | 3566 |
| Walker2d-v2 | 17 | 6 | - | 4924 |
| Ant-v2 | 111 | 8 | 6000 | 6157 |
| HalfCheetah-v2 | 17 | 6 | 4800 | 12294 |
| Humanoid-v2 | 376 | 17 | 1000 | 5187 |

The exact performance of all methods are list in Table 7, 8, 9, 10, 11, 12. We compare GAIL(Ho & Ermon, 2016), behavior cloning, GAIL with behavior cloning initilization and AIRL to our method containing. Means and standard deviations are calculated from 20 trajectories after the agents converge and the number total interactions with environments is less than one million environment steps.

Table 7: Performance on Swimmer-v2 with different trajectories

| #Demo | 5 | 10 | 20 | 50 |
|---|---|---|---|---|
| | Swimmer-v2 | | | |
| Expert | 332.88 ± 1.24 | | | |
| BC | 328.85 ± 2.26 | 331.17 ± 2.4 | 332.17 ± 2.4 | 330.65 ± 2.42 |
| GAIL | 304.64 ± 3.16 | 271.59 ± 11.77 | 56.16 ± 5.99 | 246.73 ± 5.76 |
| BC-GAIL | 313.80 ± 3.42 | 326.58 ± 7.87 | 294.93 ± 12.21 | 315.68 ± 9.99 |
| AIRL | 332.11 ± 2.57 | 338.43 ± 3.65 | 335.67 ± 2.72 | **340.08 ± 2.70** |
| Our init | **332.36 ± 3.62** | 335.78 ± 0.34 | **336.23 ± 2.53** | 334.03 ± 2.11 |
| Our final | 332.22 ± 3.23 | **339.67 ± 3.21** | 336.18 ± 1.87 | 336.31 ± 3.20 |

Table 8: Performance on Hopper-v2 with different trajectories

| #Demo | 5 | 10 | 20 | 50 |
|---|---|---|---|---|
| | Hopper-v2 | | | |
| Expert | 3566 ± 1.24 | | | |
| BC | 1471.40 ± 637.25 | 1318.76 ± 804.36 | 1282.46 ± 772.24 | 3525.87 ± 160.74 |
| GAIL | 3300.32 ± 331.61 | 3372.66 ± 130.75 | 3201.97 ± 295.27 | 3363.97 ± 262.77 |
| BC-GAIL | **3122.23 ± 358.65** | 3132.11 ± 520.65 | 3111.42 ± 414.28 | 3130.82 ± 554.54 |
| AIRL | 4.12 ± 0.01 | 3.07 ± 0.02 | 4.11 ± 0.01 | 3.31 ± 0.02 |
| Our init | 2322.49 ± 300.93 | 3412.58 ± 450.97 | 3314.03 ± 310.32 | 3601.16 ± 300.14 |
| Our final | 3092.26 ± 670.72 | **3539.56 ± 130.36** | **3516.81 ± 280.98** | **3610.19 ± 150.74** |

Table 9: Performance on Walker2d-v2 with different trajectories

| #Demo | 5 | 10 | 20 | 50 |
|---|---|---|---|---|
| | Walker2d-v2 | | | |
| Expert | 5070.97 ± 209.19 | | | |
| BC | 1617.34 ± 693.63 | **4425.50 ± 930.62** | 4689.30 ± 372.33 | **4796.24 ± 490.05** |
| GAIL | 1307.21 ± 388.55 | 692.16 ± 145.34 | 1991.58 ± 446.66 | 751.21 ± 150.18 |
| BC-GAIL | **3454.91 ± 792.40** | 2094.68 ± 1425.05 | 3482.31 ± 828.21 | 2896.50 ± 828.18 |
| AIRL | -7.13 ± 0.11 | -7.39 ± 0.09 | -3.74 ± 0.13 | -4.64 ± 0.09 |
| Our init | 1859.10 ± 720.44 | 2038.90 ± 260.78 | 4509.82 ± 1470.65 | 4757.58 ± 880.45 |
| Our final | 2681.20 ± 530.67 | 3764.14 ± 470.01 | **4778.82 ± 760.34** | 4780.73 ± 360.66 |

Table 10: Performance on Ant-v2 with different trajectories

| #Demo | 5 | 10 | 20 | 50 |
|---|---|---|---|---|
| | Ant-v2 | | | |
| Expert | 6190.90 ± 254.18 | | | |
| BC | **3958.20 ± 661.28** | 3948.88 ± 753.41 | 5424.01 ± 473.05 | 5852.79 ± 572.97 |
| GAIL | 340.02 ± 59.02 | 335.25 ± 89.19 | 314.35 ± 52.13 | 284.18 ± 32.40 |
| BC-GAIL | -1081.30 ± 673.65 | -1177.27 ± 618.67 | -13618.45 ± 4237.79 | -1166.16 ± 1246.79 |
| AIRL | -839.32 ± -301.54 | -386.43 ± 156.98 | -586.07 ± 145.43 | -393.90 ± 145.13 |
| Our init | 1150.82 ± 200.87 | 3015.43 ± 300.70 | 5200.58 ± 870.74 | 5849.88 ± 890.56 |
| Our final | 1693.59 ± 350.74 | **3983.34 ± 250.99** | **5980.37 ± 420.16** | **5988.65 ± 470.03** |

Table 11: Performance on HalfCheetah-v2 with different trajectories

| #Demo | 5 | 10 | 20 | 50 |
|---|---|---|---|---|
| | HalfCheetah-v2 | | | |
| Expert | 12294.22 ± 208.41 | | | |
| BC | 225.42 ± 147.16 | 971.42 ± 249.62 | 2782.76 ± 959.67 | 4813.20 ± 1949.26 |
| GAIL | -84.92 ± 43.29 | 474.42 ± 389.30 | -116.70 ± 34.14 | -175.83 ± 26.76 |
| BC-GAIL | 1362.59 ± 1255.57 | 578.85 ± 934.34 | 3744.32 ± 1471.90 | 1597.51 ± 1173.93 |
| AIRL | **782.36 ± 48.98** | -146.46 ± 23.57 | 1437.25 ± 25.45 | 755.46 ± 10.92 |
| Our init | 267.71 ± 90.38 | 1064.44 ± 227.32 | 3200.80 ± 520.04 | 7102.74 ± 910.54 |
| Our final | 513.66 ± 15.31 | **1616.34 ± 180.76** | **6059.27 ± 344.41** | **8817.32 ± 860.55** |

Table 12: Performance on Humanoid-v2 with different trajectories

| Humanoid-v2 | | | | |
|---|---|---|---|---|
| #Demo | 5 | 10 | 20 | 50 |
| Expert | $5286.21 \pm 145.98$ | | | |
| BC | **1521.55$\pm$ 272.14** | **3491.07$\pm$ 518.64** | 4686.05 $\pm$355.74 | 4746.88 $\pm$605.61 |
| GAIL | 485.92$\pm$ 27.59 | 486.44 $\pm$27.18 | 477.15$\pm$ 22.07 | 481.14$\pm$ 24.37 |
| BC-GAIL | 363.68 $\pm$44.44 | 410.03 $\pm$33.07 | 487.99$\pm$ 30.77 | 464.91 $\pm$33.21 |
| AIRL | $79.72 \pm 4.27$ | $87.15 \pm 5.01$ | $-1293.86 \pm 10.70$ | $84.84 \pm 6.46$ |
| Our init | $452.31 \pm 190.12$ | $1517.63 \pm 110.45$ | $4610.25 \pm 2750.86$ | $4776.83 \pm 1320.46$ |
| Our final | $1225.58 \pm 210.88$ | $2190.43 \pm 280.18$ | **4716.91 $\pm$680.29** | **4780.07 $\pm$ 700.01** |

## C  HYPER-PARAMETER AND NETWORK ARCHITECTURE

When we pretrain the policy network with our methods, we choose $\beta = 0.05$ in $\beta$-VAE. We use Adam with learning rate 3e-4 as the basic optimization algorithms for all the experiments. The policy network and value network used in the algorithms all use a three-layer relu network with hidden size 256. We choose $\sigma = 0.1$ in the policy prior for all the environments.

## D  COMPARISON WITH AIRL (FU ET AL.) FROM A THEORETICAL PERSPECTIVE

Here we illustrate the theoretical advantage of our SAIL algorithm over AIRL in certain scenarios by an example.

The theory of AIRL shows that it is able to recover the groundtruth reward of an MDP up to a constant if the reward of this MDP is define on states only, when the adversarial learning reaches the equilibrium. Next we show a basic case that violates the theoretical assumption of AIRL but can be solved by our algorithm.

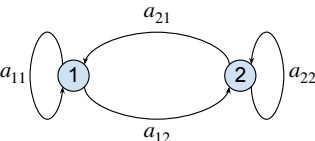

Figure 6: Two-ring MDP with deterministic transition

Figure 6 shows the states and transition of an MDP. The demonstration policy jumps back and forth between $s_1$ and $s_2$ periodically. Because our algorithm has the action prior (local alignment), it is clear that we can solve this problem. The dynamics of many periodic games, such as Walker and HalfCheetah in MuJoco, are extension of this two-ring graph.

It is easy to show that it is impossible for the adversarial game in AIRL to solve this problem at equilibrium. According to Sec 6 of Fu et al., the reward family of AIRL is parameterized as

$$f_\theta(s, s') = g(s) + \gamma h(s') - h(s) \tag{8}$$

For simplicity of notation, let $\phi(s) = g(s) - h(s)$ and $\psi(s) = \gamma h(s)$, then

$$f_\theta(s, s') = \phi(s) + \psi(s') \tag{9}$$

In other words, the reward of AIRL is decomposible to the sum of two functions defined on states only.

Again, for simplicity, we omit the arguments of functions but use subscripts to represent states. For example, $f_{12} = f(s_1, s_2)$ and $\phi_1 = \phi(s_1)$. Then,

$$\begin{aligned} f_{12} = \phi_1 + \psi_2, \quad f_{11} = \phi_1 + \psi_1 \\ f_{21} = \phi_2 + \psi_1, \quad f_{22} = \phi_2 + \psi_2 \end{aligned} \tag{10}$$

Assume that AIRL has reached the equilibrium and learned the optimal policy, then it must be true that $f_{12} > f_{11}$ and $f_{21} > f_{22}$ (otherwise, there exists other optimal policies). But $f_{12} > f_{11}$ implies that $\psi_2 > \psi_1$, while $f_{21} > f_{22}$ implies that $\psi_1 > \psi_2$, which is a contradiction.

