# OpenReview forum: "State Alignment-based Imitation Learning"
_ICLR.cc/2020/Conference — Accept (Poster)_

### Official Review · AnonReviewer1 · 2019-10-22
**Official Blind Review #1**

**Rating:** 3

**Review:**

Summary of claims:

The paper proposes an imitation learning method that aims to align state distributions rather than state-action distributions to account for cases where the imitator dynamics differ from expert dynamics. They achieve this by two objectives: one local, the other global. The local objective aligns the next state to be close to the expert's next state in each transition by first training a VAE on the expert demonstrations, and using the trained VAE in conjunction with a pretrained inverse dynamics model to compute the action that the imitator needs to imitate. The global objective tries to do a global alignment of states encountered in the imitator and expert trajectories, by minimizing the Wasserstein distance between the two trajectory distributions. The paper claims that using these two objectives results in a method that outperforms existing inverse reinforcement learning and behavior cloning approaches in settings where the imitator and expert dynamics differ.

Decision:

I recommend the paper to be rejected. I have three main reasons for my decision (with more details in the next section):
1. The paper is very poorly written : A lot of details are missing in the paper, notation is not standardized,  related work is just a list of previous papers without any context on how the proposed method is related, previous methods are referred to without any citations, and quite a few blanket statements which are not substantiated.
2. Incomplete approach description: Quite a few components of the approach are not explained (or even discussed), no intuition provided for the choices made in the approach, the concept of different dynamics is not formalized, some technical inconsistencies in the algorithm, no formal problem statement (which would really help in standardizing notation), and most claims made about the approach are not justified or substantiated
3. Poor experiments: Experiments are not well chosen to reflect the premise and claims of the paper, little to no details given for how the baseline approaches were trained, no details on policy parameterization, and missing comparison with baseline approaches in some experiments

Comments:

(1) Problem setup:
(a)Problem setup is very vague and not formalized.
(b) Differing dynamics could mean several things: different agent dynamics (like different action spaces; different actuators; etc.), different environment dynamics (different moving obstacles in the world;) etc.
(c) Basically, different dynamics can mean a lot more than what was accounted for in the paper

(2) Blanket Statements:
(a) The authors keep saying that their framework is more flexible without any justification as to why,
(b) "simply train an inverse dynamics model"- training an inverse dynamics model can be very hard especially when environment dynamics are stochastic/when the inverse model is multimodal,
(c) "constraint becomes loosened" - this statement doesn't make any sense without more explanation,
(d) Several other blanket statements about existing approaches

(3) Notation :
(a) Notation was never standardized in the paper,
(b) what is \phi? what is the input-output of \phi?
(c) What is \theta_old? What is \sigma?
(d) There are a lot of things that needed explanation, especially in the algorithm

(4) Related Work :
(a) The related work section is just a dump of citations without giving any context for where the proposed work lies in the spectrum of these works. How does it compare? Why is it better/worse?
(b) Missing related work that was publshed in ICML 2019 that has a very similar approach in matching state distributions ("Provably efficient Imitation Learning from Observations Alone" or FAIL) and works very well.

(5) Motivation :
(a) The approach, in general, needs better motivation. The toy example in the introduction was good but the experiments did not reflect the complexity of that example.
(b) Try to have a running example in the paper that will help you motivate the approach better.

(6) Background :
(a) The background section is very minimal and lacks any details necessary.
(b) I had to read the beta-VAE paper to understand what it does.
(c) The section also lacks any minimal background in IL/RL and the notation could also have been standardized in this section

(7) Figures:
(a) All the figures in this paper could use a lot of improvement in terms of descriptive captions, more informative legends, bigger fonts, descriptive text, and more figures as well

(8) Algorithm :
(a) The algorithm was not referenced anywhere in the text,
(b) No definition for \tau, details on pretraining inverse dynamics model lacking in both text and algorithm,
(c) *Policy prior is used to pretrain policy before the VAE was trained!* which doesn't make any sense since the policy prior is obtained using the VAE,
(d) the equation at the end of Sec 4.3 has r(s, a) whereas reward is defined as r(s_t, s_{t+1}) but they are not equivalent when dynamics are stochastic

(9) Experiments:
(a) What is AIRL? No reference was given.
(b) Why is keeping the variance of the policy constant reasonable? How do you come up with the value?
(c) How do you pretrain the VAE, invserse dynamics model?
(d) The setup of making the ant's legs smaller or body heavier seems very artificial. I am sure its easy to come up with more realistic setups in navigation domains, for example. Try to use more realistic experiments in the future.
(e) For results in Fig 3, is AIRL, GAIL also pretrained with VAE or by BC? Seems like SAIL was pretrained but the others weren't since SAIL starts off with a high score at the start.
(f) For Sec 5.1.2, comparison with baseline approaches are missing. Legends for the plots are terribly small.

Conceptual questions:
1. How do you account for cases where due to differing dynamics, states reached by expert in demonstrations are unreachable by the imitator?
2. How do you account for cases where the environment dynamics changes between expert and imitator?
3. Why does using Wasserstein distance make sense? Why not other f-divergences? Also, matching global distributions can be very misleading if you have states that are visited multiple times in the same trajectory. FAIL recommends matching state distribution at each time-step instead and is much more stable
4. How is this different from GAIL where we match state visitation distribution (instead of state-action visitation distribution?)

Things to improve:
1. Writing needs to be improved a lot
2. Better experiments - more realistic domains
3. Approach needs to be explained more formally

**Experience Assessment:**

I have published one or two papers in this area.

**Review Assessment: Checking Correctness Of Derivations And Theory:**

I carefully checked the derivations and theory.

**Review Assessment: Checking Correctness Of Experiments:**

I assessed the sensibility of the experiments.

**Review Assessment: Thoroughness In Paper Reading:**

I read the paper at least twice and used my best judgement in assessing the paper.

---

> ### Author Response · Authors · 2019-11-14
> **Response to Reviewer#1**
>
> Q1: The paper is very poorly written
> A: We agree that some minor details are omitted due to space limitations. This paper, like many other accepted ICLR papers, is built upon a rich set of techniques. Our basic building blocks,  VAE (and beta-VAE), Wasserstein Distance (and its duality), and PPO are very popular recently, so we assumed familiarity of readers in the ML community. While we have to be succinct in explaining them to save space, our presentation (including the choices of notations and symbols) has respected the convention by choosing the most common naming. Nonetheless, we would like to polish the writing with more details and update the Method Section, including the algorithm. We also plan to release the code to help the understanding of method details.
>
> In fact, R3 complimented that our paper is generally well-written and can be followed, and R2 also seems to have followed our presentation by recommending acceptance.
>
>
>
> Q2: some notations (\phi, \theta_old, \sigma)
> A: $\phi$ in Eq. 3 is the Kontorovich potential of Wasserstein Distance, which is also the discriminator in the WGAN paper, as mentioned in the context of Sec 4.3 of our submission. $\theta_old$ is the parameter for the policy network in PPO, following the convention of the original PPO paper, as mentioned in the context of Eq. 6. $\sigma$ in Eq. 7 is the variance of the action prior, and $\sigma$ in Sec 4.5 is the variance of the Gaussian policy. We will explain the symbols clearly in the revision.
>
> Q3: Different dynamics can mean a lot more than what was accounted for in the paper. Different agent dynamics versus different environment dynamics. Problem setup vague
> A: We have clearly stated our problem setup in the abstract (opening sentence) and the introduction, that we only care about different *agent dynamics*.
>
> Q4: Blanket statements
> A: “flexible”: We say our method is more flexible because one does not have to guarantee that the demonstrator and the imitator have exactly the same dynamics when using it.
>
> "constraint becomes loosened": GAIL-based methods require to match the distribution of state-action *pairs*, whereas the global alignment component in our method only requires to match the distribution of single states (the second paragraph of Sec 4.3). It is clear that the constraints are loosened: When the state-action matching is achieved, the state matching will also be achieved, but the inverse direction may not hold.
>
> “simply train an inverse dynamics model”: Training an inverse dynamics model can indeed be very hard, but it is not that hard if we have access to the low-dimensional control states and the dynamics are non-stochastic, as in this work. Here, “simply train” just means that we train a plain neural network without any tricks. In the revision, we would like to adjust our presentation and be precise about the statement.
>
> Q5: Motivation. The toy example in the introduction was good but the experiments did not reflect the complexity of that example, ...
> A: In Sec 5.1 of our experiments, one of the physical properties that we adjusted is the mass density of agents (stated in Sec 5.1.1), which influenced agent moving speed as in our toy example. Specifically, the heavier agent will run slower given the same force and torque. We showed results in Ant and Swimmer environments.
>
> Q6: Background (“very minimal”, “I have to read the beta-VAE paper”, “lacking any minimal background in IL/RL”)
> A: We did not include too many details in the background section due to the limit of space. And we feel so sorry if it causes any difficulty in understanding. However, other reviewers also asked us to include more experiments and in-depth analysis, so we have to discard some common knowledge in RL/IL community.
>
> Q6: Figures
> A: We will improve the font size and captions in the updated version.
>
> Q7: Algorithm
> A: Thanks for your proofreading. \tau is the trajectory of the agent. The 6 and 7 lines should be reverted. And the reward is defined on states, so it should be r(s, s’). We will update these points during the rebuttal period.
>
> Q8: Experiments
> - AIRL is the paper ‘LEARNING ROBUST REWARDS WITH ADVERSARIAL INVERSE REINFORCEMENT LEARNING’. We will explicitly define this acronym.
> - For the policy prior, the variance is a hyper-parameter. We use 0.1 in our experiments. The policy network is initialized to have the same variance as the policy prior and then adjusted online, so it’s not a constant during the interactions.
> - We train VAE using the demonstrations. The input to VAE is s_t and the supervision is s_{t+1}. We train the inverse dynamics using the rules described in the first 5 lines of the algorithm pseudo-code.
> - All MuJoCo environments have artificial models. But it’s still worthwhile to test an algorithm on that given the massive publications in ICLR/ICML/NeurIPS that only test in simulators.
> - For Sec 5.1.2, as the two agents have different action dimensions, the baseline methods cannot complete this task.

---

> > ### Author Response · Authors · 2019-11-14
> > **[cont'd]**
> >
> > - In Fig 3, the other methods are not pre-trained with BC because we already listed BC as a baseline. In Sec 5.2 (actors of the same dynamics), we showed the performance of GAIL pre-trained with BC, which did not outperform BC in general (Table 2, Table 7-12). In practice, while the pre-trained policy network performs reasonably well at initialization, the policy update in GAIL is likely to worsen the initial policy, especially when the demonstration is abundant. The observation leads us to believe that the performance of pure inverse RL methods (e.g., GAIL and AIRL) at equilibrium cannot be simply improved by BC pretraining. Therefore, we didn’t compare GAIL+BC or AIRL+BC in other settings.
> >
> > Conceptual questions:
> > Q1. How do you account for cases where due to differing dynamics, states reached by expert in demonstrations are unreachable by the imitator?
> > A: The disabled ant experiment empirically shows the effectiveness of our method in such cases. While the disabled ant is not able to move as steadily as the demonstrator because some states are unreachable, it still roughly follows the trajectory of the demonstrator.
> >
> > If the VAE generated an unreachable state $s’$, then the inverse dynamics model would be likely to output an action that leads to a state $s’’$ close to $s’$. This is a result of the smoothness of the inverse dynamics model (a neural network) and the environment dynamics (in most cases it is true). Additionally, our global alignment tends to push the imitator back to familiar states. The global and local alignment mechanisms together provide certain fault tolerance.
> >
> > Q2. How do you account for cases where the environment dynamics changes between expert and imitator?
> > A: This is not the case we study in this paper, clearly mentioned in the opening sentence of our abstract. Besides, we feel that expert and imitator should refer to agents, not environments.
> >
> > Q3. Why does using Wasserstein distance make sense? Why not other f-divergences?
> >
> > A: While the choice of distribution discrepancy measure is really not the focus of the work, we provide the argument why we choose Wasserstein distance but not f-divergence: Wasserstein distance is widely used in fields such as generative adversarial networks (WGAN by Arjovsky et al.) and image retrieval (EMD distance by Rubner et al.). Compared with KL divergence (and its variants f-divergence), Wasserstein distance has many nice theoretical and numerical properties:
> > - Wasserstein distance allows us to compare the discrepancy between distributions of a broader family. When two distributions have negligible intersections, computing f-divergences will have trouble (not defined or simply infinite). This situation is quite common for matching high dimensional state space. More explanations and examples can be found in the WGAN paper.
> > - Optimizing Wasserstein distance is numerically more stable than f-divergence (the WGAN paper gives examples that f-divergence such KL-divergence cannot optimize).
> > - Wasserstein distance is a proper metric and is suitable for continuous interpolation of distributions, as shown in paper [Convolutional Wasserstein Distances, SIGGRAPH2015 by Solomon et al]. In our setting, we also want the distribution of visited states by the imitator to gradually move towards the demonstration distribution.
> >
> > Q4. matching global distributions can be very misleading if you have states that are visited multiple times in the same trajectory. FAIL recommends matching state distribution at each time-step instead and is much more stable
> > A: Only matching global states will be misleading. However, the local alignment of VAE will resolve this issue (explained in Sec 4.3 of our paper). We recently also compared our method with FAIL. The released code of FAIL only supports discrete actions. The FAIL paper claimed that it can be extended to continuous action space and we accordingly modified the FAIL code so that it can support continuous control. However, the continous FAIL cannot work well in the standard imitation learning setting (same expert/imitator dynamics) for Swimmer. Continuous FAIL reached 120 points, while ours can reach over 300 points. On the standard HalfCheetah imitation learning setting , FAIL can not achieve positive points, while our method can achieve almost 9000 points.
> >
> > Q5. How is this different from GAIL where we match state visitation distribution (instead of state-action visitation distribution?)
> > A: Compared to the state-based GAIL, we have a local alignment component to make sure the relationships between consecutive states are preserved. This is necessary because matching global distribution alone would be misleading, as you said.

---

> > > ### Comment · AnonReviewer1 · 2019-11-15
> > > **Response to Authors and Final Decision**
> > >
> > > Thanks for your descriptive reply!
> > >
> > > About Q1, while I agree that page limit restricts the space you could have spent describing background approaches, I strongly feel that notation needs to be standardized in each paper. VAE/PPO/WGAN are not papers that every IL/RL researcher needs to read and refer to while reading your paper to understand the notation. Expecting the reader to put in a lot of effort understanding your work highlights that the writing is lacking.
> > >
> > > Q2, I would like to see the notation introduced in a future manuscript.
> > >
> > > Q3, I just did a word search on the manuscript and there is no phrase "agent dynamics" anywhere in the paper. What the abstract and introduction do say is that the expert and imitator have different dynamics models, which as I said could mean both environment/agent dynamics. Please be specific in a technical paper.
> > >
> > > Q4, please include the explanations given in the rebuttal in the future manuscript.
> > >
> > > Q5, I don't think unless you quantify how different the dynamics are for a lighter and heavier ant by providing trajectory statistics, you can make a statement that your approach can deal with differing agent dynamics. I personally liked your toy example much more than the experiments done in the paper. Please choose an experimental domain that clearly highlights a need for an approach that is robust to changes in agent dynamics and show how your approach solves the challenge.
> > >
> > > Q6, defer to Q1
> > >
> > > Q7, update the manuscript as you described
> > >
> > > Q8, please provide the details in the rebuttal in the paper as well
> > >
> > > Given the current state of the paper and the details provided in the rebuttal, I am going to change my decision to a weak reject but can't improve it any further. The authors should seriously consider a more challenging realistic domain that showcases the strengths of their approach. In addition to this, the authors should spend more time on writing the paper to make sure its as self-contained as possible (to the extent where readers don't have to read other papers to understand notation.)
> > >
> > > Thanks for spending the time writing the rebuttal and I hope you keep up the good work, irrespective of the final decision. :)

---

### Official Review · AnonReviewer2 · 2019-10-23
**Official Blind Review #2**

**Rating:** 8

**Review:**

This paper seeks a solution to the problem of performing imitation learning when the dynamics of the demonstrator are different from the dynamics of the imitator. The authors present a novel approach that combines global alignment by minimizing the Wasserstein distance between state occupancies with local alignment via a state-predictive VAE and inverse dynamics model. The experimental results support the claims that the method works for different dynamics and the proposed approach usually outperforms existing imitation learning methods.

The problem of dealing with different dynamics between a demonstrator and imitator is an important, but often overlooked problem in imitation learning. The combination of the global and local alignment is novel, nicely motivated, and ablation studies demonstrate that both are needed for good performance. Given the extensive experimental results showing the efficacy of this method I recommend that the paper be accepted.

However, I feel that the paper can still be improved. Below are some of my questions and suggestions.

The success of BC is interesting. Why does it do so well? This seems to violate the motivation of the paper that using (s,a) for imitation learning won't work if the dynamics change.

I thought the experiments for different action dynamics was very nice. The paper mentions that even state-spaces cannot be matched between the point mass and Ant. How do you know what part of the state space to imitate? Do you assume that is prior knowledge? Is this something that could be learned or inferred? How?

How is the potential updated. Eq(3) doesn't give an update rule. From later discussion it appears you mean take the \phi that results in the supremum. Is this value learned? How is the optimization performed to solve Eq(3)?

How would you make the policy prior in Eq(7) work if actions are discrete? What if actions are multidimensional, with different ranges where some actions are not important? What is sigma?

In the RL community there is interest in transfer learning when the dynamics change. If the reward were observable, would the current approach be potentially useful for boosting the performance of transfer learning in RL?

Related Work:

The authors cite AIRL, but could do a better job distinguishing between AIRL and the current work. AIRL also tries to learn a state-based reward that is disentangled from the dynamics. Are there theoretical reasons why this work is better? Why does the proposed method work so much better in practice. On a related note, the results for the disabled ant seem much lower than those presented in the original AIRL paper. Why is this?

The authors do not mention the work by Brown et al. "Extrapolating Beyond Suboptimal Demonstrations via Inverse Reinforcement Learning from Observations", ICML 2019. This work also learns only from observations and does not require any pretraining. How is the current work different?

The authors cite the work by Ayatar on imitation learning from observing YouTube videos. This method is also state-based and uses a similar state occupancy matching reward that would also work with different dynamics. How is the current method different. Would the current method work on domains such as learning to play Atari from raw visual trajectories?

Typos:

"... able to resume to the demonstration trajectory by itself."
--maybe say " able to return to the demonstration trajectory by itself."

"... pairs as in an observation-based GAIL (Ho and Ermon 2016). I think this should be Torabi et al. instead.

Page 5 "state predictive VAE and an inverse *dynamics*"




**Experience Assessment:**

I have published in this field for several years.

**Review Assessment: Checking Correctness Of Derivations And Theory:**

I assessed the sensibility of the derivations and theory.

**Review Assessment: Checking Correctness Of Experiments:**

I carefully checked the experiments.

**Review Assessment: Thoroughness In Paper Reading:**

I read the paper thoroughly.

---

> ### Author Response · Authors · 2019-11-13
> **Response to Reviewer#2**
>
> Thank you for your encouraging and constructive comments!
>
> Q1: The success of BC is interesting. Why does it do so well? This seems to violate the motivation of the paper
> A: It seems like BC is more robust to dynamics changing compared with GAIL and AIRL. But the motivation of our paper is not really violated based on the following experiment observations: In the experiment of Sec 5.1.1, we find that BC will cause disabled ant and heavy ant to reach dangerous states by cloning actions (the ant would flip over with stomach upwards), yet our state-based imitation method will not approach these states. So we can outperform BC by thousands of scores (~2000 and ~5500 respectively). However, in the other four games, BC agents do not die but just move less efficiently. Benefited from the (limited) network extrapolation ability, the agents can get sub-optimal but still reasonable scores.
>
>
> Q2: I thought the experiments for different action dynamics was very nice. Do you assume that is prior knowledge? Is this something that could be learned or inferred? How?
> A: We use the same setting as many hierarchical RL papers, i.e, the agent need to reach a goal position in a maze, which is represented by (x,y) coordinates. We also know that the first two dimensions of states are the position of the agent. The prior knowledge includes: (1) the goal space (or the common space that need to be matched) (2) the projection from the observation to the goal space (select the first two dimensions of the states). However, this can also be learned, we can adopt some metric learning methods to match states from different agents. We leave it as future work.
>
> Q3: How is the potential updated. Eq(3) doesn't give an update rule. From later discussion it appears you mean take the \phi that results in the supremum. Is this value learned? How is the optimization performed to solve Eq(3)?
> A: This \phi is a neural network, so we can optimize its parameters by maximizing the objective in Eq (3). This is also the max-game in Wasserstein GAN (serves as WGAN’s discriminator), and we use gradient penalty to ensure the Lipschitz of the network (as in WGAN-GP).
>
> Q4: How would you make the policy prior in Eq(7) work if actions are discrete? What if actions are multidimensional, with different ranges where some actions are not important? What is sigma?
> A:  When actions are discrete, we can still use a network with softmax layer to predict the action distribution from state transition (s, s’). Then we can plug this action distribution into the KL divergence term in Eq (5) to impose a prior over actions.
> The sigma in Eq (7) controls how strong the action prior is when regularizing the policy update.
> In our experiments with multidimensional actions, we found that setting the same sigma for all the dimensions is fine. When the ranges on different action dimensions differ significantly, a possible solution would be to scale the sigma of each dimension based on the variance estimated from demonstration actions.
>
> Q5: In the RL community there is interest in transfer learning when the dynamics change. If the reward were observable, would the current approach be potentially useful for boosting the performance of transfer learning in RL?
> A: We feel that this is a very interesting future direction! For transfer learning in RL, the fundamental question is still what are shared across environments. Our results show that, when only the actor dynamics are changed, distilling information from the state visitation trajectory is a promising idea. Indeed, in the transferrable RL setting, we may set the observed reward as an extrinsic reward, our reward from Wasserstein state matching as an intrinsic reward, and our VAE-based prior from inverse dynamics as the action prior.
>
> Q6: The authors cite AIRL, but could do a better job distinguishing between AIRL and the current work. AIRL also tries to learn a state-based reward that is disentangled from the dynamics. Are there theoretical reasons why this work is better? Why does the proposed method work so much better in practice. On a related note, the results for the disabled ant seem much lower than those presented in the original AIRL paper. Why is this?
>
> A: We sincerely thank you for the insightful perspective. There indeed exist theoretical reasons that our approach may work better than AIRL in certain cases. We have created a PDF under an anonymous link to illustrate this point: https://drive.google.com/file/d/1yqwYfLuVg0gw61gVu7g91sxD_cNPI4u0/view?usp=sharing. About AIRL’s performance difference of disabled ant in our paper versus in the original paper, it is because our environment is based on the ant model provided in OpenAI gym, but AIRL uses rllab. Additionally, we kept the original control range unchanged [-150, 150], while AIRL mitigated the control range to [-30, 30] of the ant to make it more stable.

---

> > ### Author Response · Authors · 2019-11-13
> > **Response to Reviewer#2 [cont'd]**
> >
> > Q7: The authors do not mention the work by Brown et al. "Extrapolating Beyond Suboptimal Demonstrations via Inverse Reinforcement Learning from Observations", ICML 2019. This work also learns only from observations and does not require any pre-training. How is the current work different?
> >
> > A: Although this work uses state-based IRL, their setting is quite different from ours. They do not focus on the case when the dynamics are different. Instead, the purpose of this paper is to extrapolate beyond the sub-optimal demonstrations.
> >
> > Q8: The authors cite the work by Ayatar on imitation learning from observing YouTube videos. This method is also state-based and uses a similar state occupancy matching reward that would also work with different dynamics. How is the current method different. Would the current method work on domains such as learning to play Atari from raw visual trajectories?
> >
> > A: The work by Ayatar et al. reduces the domain gap between YouTube video and Atari by using metric learning. They also define rewards based on the local state matching in the embedding space, so that an RL can be used to solve the task. This work shares similar spirit as ours that both works are state-based. However, their core challenge lies in how to bridge the domain gap between YouTube and Atari visual appearances, yet its reward design is much easier since the action space in Atari games is quite limited, in comparison to our continuous and high-dimensional action space. By our experiments, the reward function in Sec 4 by Ayatar et al. is not feasible in solving our MuJoCo imitation learning tasks.
> >
> > We feel that our approach should be able to play Atari games from raw visual trajectories. If time permits, we would give a try.

---

> > > ### Comment · AnonReviewer2 · 2019-11-14
> > > **Response to authors**
> > >
> > > I thank the authors for their detailed response. I encourage the authors to revise their draft for clarity and include some of the discussion in the responses to the reviews in the appendix. I think the example of where AIRL fails is also nice and would be a good addition to the appendix.
> > >
> > > The authors have addressed all of my main concerns. I think this paper will be of interest to many in the imitation learning and reinforcement learning communities. Based on the novelty of the approach, the importance of imitation learning with different agent dynamics, and the significant improvement over state-of-the-art, I think the paper should be accepted into ICLR.

---

> > > > ### Author Response · Authors · 2019-11-15
> > > > **A short update on Atari environments**
> > > >
> > > > We have tried our method on Atari and now update the result here. Due to the time limit, we only tested our method on Pong. And in Atari experiments, the expert and the imitator have the same dynamics model because it is very hard to modify the Atari game at the engine level. Since computing Wasserstein distance on image domain is pretty difficult, we remove the global alignment component in our method. Given 20 expert demonstration trajectories (all of them have the perfect score 21), our imitator can also achieve the average score of 21 in 30 testing episodes. Given more time, we may try to figure out a way to test the imitation with different dynamics in Atari.

---

### Official Review · AnonReviewer3 · 2019-10-29
**Official Blind Review #3**

**Rating:** 6

**Review:**

Review for "State alignment-based imitation learning."

Summary:

This paper addresses the problem of learning from demonstrations where the demonstrator may potentially have different dynamics from the learner. To address this, the paper proposes to align state-distributions (rather than state-action distributions) between the demonstrations and the learner. To alleviate issues arises from doing this alone, they also propose to use a local learned prior to guide their policy to select actions that take it back to the original demonstrations. The paper shows a number of experiments on control tasks in support of their claims.

Pros:
+ The problem that the paper solves is fairly relevant, and some experiments (such as cross morphology imitation learning) are promising in concept.
+ The paper is mostly well written (save some small improvements that could be made in clarity) and can be followed.
+ The paper presents a series of experiments across several agents and some other baselines.

Cons (and primary concerns):

1) The idea of matching state distributions in the context of learning behaviors is not new. In particular, [1] also uses similar ideas of matching state distributions in the imitation learning context, if via different machinery. [4] points towards such ideas as well (noted on page 43). Works such as [2, 3] also use this idea in the context of Reinforcement Learning. Further, ideas of deviation correction in the imitation learning domain have been addressed before in [5]. The paper would benefit from a more thorough treatment of these related works, and how the proposed work differs from these.

2) The choice of approach (in particular, the use of the Wasserstein distance to match state distributions, and the manner of learning a local prior by training an autoregressive Beta VAE) are lacking motivation, and it is unclear if or why these choices are the best way to approach the problem.

3) While the paper presents a large number of comparisons, the analysis of the relative performance of the proposed approach against the baselines is lacking. For example, in section 5.1.1., vanilla BC seems to do very well - why is it the proposed approach only marginally outperforms BC on several of these tasks? In section 5.2, why is SAIL able to outperform other IL techniques on same-dynamics tasks? What about SAIL provides this performance benefit? Similarly, in section 5.3.3, what is it about the Wasserstein objective and the KL that together enables good learning? This ablation seems crucial to assessing the paper, and is lacking a deeper analysis. Further, the relevance of section 5.3.1 is questionable - as no new insight is provided over the original Beta-VAE paper.

Other Concerns:

1) The paper ultimately uses a form of prior that is defined over actions, and not states (so that it may be used in the KL divergence term). How is the choice of form of prior made? It is unclear why it is better to have a prior learned over states converted to actions via eq. 7, versus a similarly designed prior over actions.

2) It is unclear why the expression of the reward function (Eq. 4) is necessary - if it is possible to compute the Wasserstein distance (and hence the cummulative reward), it is possible to update the policy purely from this cummulative reward.
Is the function of the per-timestep reward simply to provide a denser signal to the policy optimization?

3) The authors claim to introduce a "unified RL framework" in their regularized policy objective. It appears that this is simply the addition of the KL between the policy and the prior $p_a$ to the global alignment objective (subsumed into $L_{CLIP}$), hence the reviewer questions whether this can indeed be treated as a novel contribution of the paper.

4) The problem this paper addresses (and the fundamental thesis for its approach) is that action-predictive methods are likely to suffer from deviation from the original demonstrations, as compared to state-predictive methods.
What purpose does section 5.3.2 serve beyond reiterating this point?

5) State-matching (and the implied use of inverse models) means the feasibility of retrieved actions is not guaranteed, as compared to models that predict actions directly.

Minor Points:

1) Explaining the various phases of training (as observed in the algorithm) would be useful.

2) Discussing how states are compared in the cross morphology experiments (Section 5.1.2.) would also be useful.

Cited Literature:

[1] State Aware Imitation Learning, Yannick Shroecker and Charles Isbell, https://papers.nips.cc/paper/6884-state-aware-imitation-learning.pdf
[2] Efficient Exploration via State Marginal Matching, Lisa Lee et. al., https://arxiv.org/abs/1906.05274
[3] State Marginal Matching With Mixtures Of Policies, Lisa Lee et. al., https://spirl.info/2019/camera-ready/spirl_camera-ready_25.pdf
[4] An Algorithmic Perspective on Imitation Learning, Takayuki Osa et. al., https://arxiv.org/pdf/1811.06711.pdf
[5] Improving Multi-step Prediction of Learned Time Series Models, Arun Venkatraman et. al., https://www.ri.cmu.edu/pub_files/2015/1/Venkatraman.pdf

Initial Decision: Weak reject

#######
Post Rebuttal Comments:

Considering the authors' motivations of approach and the additional analysis provided in the comments below, I change my decision to weak accept. I would like to encourage the authors to include the details listed below in their paper.

**Experience Assessment:**

I have published in this field for several years.

**Review Assessment: Checking Correctness Of Derivations And Theory:**

I assessed the sensibility of the derivations and theory.

**Review Assessment: Checking Correctness Of Experiments:**

I assessed the sensibility of the experiments.

**Review Assessment: Thoroughness In Paper Reading:**

I read the paper at least twice and used my best judgement in assessing the paper.

---

> ### Author Response · Authors · 2019-11-11
> **Response to Reviewer#3**
>
> Per your request for detailed motivation and analysis, we compose this lengthy response. Thank you for your patience!
>
> Response to major concerns:
>
> Q1 (Cons 1): The idea of matching state distributions is not new.
>
> A: We appreciate your suggestion of literature [1-5]. We feel that they are remotely relevant, but are happy to discuss them in revision, given the scarcity of literature in state-based approaches.
>
> First and foremost, our work explores *different* problem settings from [1-5]. We focus on the setting when the demonstrator and imitator have different dynamics models (cross-morphology setting), as stated in the opening sentence of our abstract. In contrast, [2, 3] (ICLR workshop papers uploaded on arXiv in June, 2019) study reinforcement learning; [4] (“P43 of a book”) mentioned the general idea of state distribution matching for imitation learning without giving a concrete algorithm; [5] works on multi-step prediction (no policy learning); and [1], being the most relevant that also studies imitation learning, does not consider the cross-morphology setting.
>
> In fact, our *core message* is that state-based approach is a strong candidate versus alternatives (e.g., state-action matching) in this specific scenario, a point which has not been raised before, to our knowledge.
>
> Second, while the idea of state-matching is not so new, we emphasize that our key idea is not really “state distribution matching”, but more exactly, “state alignment” between the state sequences from the demonstration and the imitation. We explain the differences. Matching state distribution alone, in fact, is often not feasible: Consider the Mujoco HalfCheetah environment. The agent needs to iterate over some certain states as much as possible. However, only looping a few times would still match the distribution, yielding a very low score. Our approach emphasizes “state-alignment”, which augments state distribution matching by taking relationships of two consecutive states into account with robustness concern (critical for the cross-morphology setting). This is achieved by our VAE-based local alignment prior, that favors state transition pairs (s, s’) resembling those in the demonstration. The movement from matching states to aligning states opens a new space.
>
> It is possible to try the idea in [1] using our framework. For example, we may replace our distribution matching module (Wasserstein distance-based) by directly optimizing the policy using gradient information as in [1]. We are implementing this alternative during the rebuttal period and would update you about the results.
>
> Q2 (Cons 2): The choices of approach are lacking motivation.
>
> A: We introduced VAE and Wasserstein distance in the background section without much explanation and motivation when using them, due to their great popularity in the ML community and space limitation. Here, we motivate their usage with details.
>
> VAE (beta-VAE): VAE is a popular tool in image modeling. This approach has an inherent data augmentation step by sampling in the latent space. VAE can predict robustly even when the test data is slightly off the training data manifold, thus benefiting many reconstruction tasks such as image denoising. We are hence motivated to use VAE in our work: We need a tool to robustly predict the next state based on the previous state, a state which may be off demonstration due to the cross-morphology setting. For the theoretical analysis of VAE, please refer to the paper by Dai et al. cited in our submission. Beta-VAE is a modified version of the original VAE. It adds a hyperparameter, beta, to control how much the variance would be when sampling augmented data.
>
> Wasserstein distance: This tool is widely used in fields such as generative adversarial networks (e.g., WGAN by Arjovsky et al.) and image retrieval (e.g., EMD distance by Rubner et al.). Compared with KL divergence (and its variants f-divergence), Wasserstein distance has many nice theoretical and numerical properties:
> - First, Wasserstein distance allows us to compare the discrepancy between distributions of a broader family: when two distributions have negligible intersections, computing KL family of divergences will have trouble (not defined or simply infinite). This situation is quite common for matching high dimensional state space. More explanations and examples can be found in the WGAN paper.
> - Second, optimizing Wasserstein distance is numerically more stable than KL/JS divergence (WGAN paper gives examples that KL/JS cannot optimize).
> - Third, Wasserstein distance is a proper metric and is suitable for continuous interpolation of distributions, as shown in paper [Convolutional Wasserstein Distances, SIGGRAPH2015 by Solomon et al]. In our setting, we also want the distribution of visited states by the imitator to gradually move towards the demonstration distribution.
>
> The choices we make are based on the above theoretical knowledge and our empirical evaluation.

---

> > ### Author Response · Authors · 2019-11-11
> > **Response to Reviewer#3 [cont'd]**
> >
> > Q3 (from Cons 3): The analysis of the relative performance of the proposed approach against the baselines is lacking.
> >
> > A: Due to the extensive experiments and space limit, we were not able to include enough detailed analysis at submission.
> >
> > In Sec 5.1.1, BC will cause disabled and heavy ants to reach dangerous states by cloning actions, yet our method will not approach these states by using state-based imitation. So we can outperform BC for thousands of scores (~2000 and ~6000 respectively). In the other four games, BC agents do not die but just move not so efficiently, so they have a sub-optimal yet still reasonable score.
> >
> > In Sec 5.2, the advantage of our methods over BC should be attributed to the inherent data augmentation by VAE. On Hopper-v2, we are significantly better with 10 demos but are just on par if the demos are increased to 50. On HalfCheetah-v2, the demo cheetah runs almost perfectly (~12294 scores); in other words, the demo provides limited instruction when the imitator is even slightly off the demo states, thus the robustness from VAE becomes critical.
> >
> > Q4 (from Cons 3): in Sec 5.3.3, Wasserstein objective and the KL together enable good learning?
> > A: We have already provided some explanations in the last paragraph of 5.3.3. Here we explain deeper. Wasserstein objective is used in our inverse RL based mechanism that would significantly penalize the exploration when the agent deviates from the demonstration far away. However, using this objective alone lacks constraints over consecutive states, as mentioned in our response to Q1, thus performing worst among all three curves in Fig 5. The KL objective adds constraints over consecutive states using a VAE prior; however, VAE is unable to extrapolate to states when the imitator deviates far from the demo (green line gradually fails as in Fig 5 (b)), but this is the scenario when the Wasserstein distance would not favor, thus the reward from the Wasserstein distance will push the imitator back to the demonstration states. We are glad to explain more if you can make your question clearer.
> >
> > Q5 (from Cons 3): relevance of Sec 5.3.1 is questionable
> > A: Sec 5.3.1 is to empirically show the role of beta and check the sensitivity of our algorithm with respect to beta. We feel that this experiment is necessary to convince the readers about our choice of beta VAE (over vanilla VAE).
> >
> > -----------------------
> > Responses to other concerns.
> >
> > Q1: How is the choice of the form of prior made? It is unclear why it is better to have a prior learned over states converted to actions via eq. 7, versus a similarly designed prior over actions.
> >
> > A: We hope that you can be clearer about your question. Because we consider cross-morphology imitation, we cannot directly learn a policy network to predict actions and plug it in the prior over actions (the KL term). Instead, we learn a state prediction network and use a learned inverse dynamics to decode the action. This way, we induce a prior over actions from the state prediction network.
> >
> > Q2: Is the function of the per-timestep reward simply to provide a denser signal to policy optimization?
> >
> > A: Yes, using dense rewards makes policy learning easier.
> >
> > Q3:  Whether the unified objective can be regarded as a novelty.
> > A: First of all, we are the first method to combine BC and IRL by regularized policy updates. GAIL mentioned combining the two by pretraining the policy. However, pre-training alone, which is equivalent to the initial score of our approach before optimizing the RL objective, will not work as well as our final results after the RL-based optimization. For a more in-depth analysis of why the combination of BC and IRL can enhance each other in our framework, please refer to the response to Q4 of your major concerns.
> >
> >
> > Q4: What purpose does Sec 5.3.2 serve beyond reiterating the deviation problem?
> > A: There are two factors that may contribute to the superior performance of our local alignment model: the robustness of VAE and the state-based representation. This experiment consolidates that the advantage really comes from our state-based approach but not the robustness of VAE (VAE for action prediction does not work well).
> >
> > Q5: State-matching (and the implied use of inverse models) means the feasibility of retrieved actions is not guaranteed, as compared to models that predict actions directly.
> > A: First of all, since NNs are involved, even predicting actions would not guarantee the feasibility of actions, especially in unseen states. Note that the actions from our method can be viewed as predicted by a big network that composes a state prediction network and an inverse dynamics network, both of which are trained in a supervised manner (supervision are obtained from the demonstration or random trials). In the cross-morphology imitation setting, predicting actions is actually even more infeasible than our method. In practice, we always clip the actions to make them feasible.

---

> > > ### Comment · AnonReviewer3 · 2019-11-15
> > > **Response to Authors**
> > >
> > > Thank you for your detailed response to my comments, (and apologies for not being able to respond to them earlier).
> > >
> > > I appreciate that you have provided a thorough motivation of your choices of approach here, and hope that these choices are included in the paper. Likewise, the analysis you provide in the above comments is insightful, and I hope that this analysis can be provided in the paper (with details or supporting experiments in the supplementary).
> > >
> > > Considering these details of motivation and analysis are added to the paper, I change my decision to weak accept.
> > > I am still unconvinced regarding the exact differences between state matching and state alignment - please consider making this distinction clear in the final version of the paper.

---

> > > > ### Author Response · Authors · 2019-11-15
> > > > **An Update for Experimental Comparison with State Aware Imitation Learning**
> > > >
> > > > In practice, this algorithm is computationally expensive due to the need to compute a big Jacobian matrix for each expert transitions (Eq 5 in their paper). While it has been verified on some small environments in the original paper, we find that it is not a straightforward effort to make it computationally affordable for environments in our submission.
> > > >
> > > > In the experiment, we compare this algorithm with ours in the Disabled-Swimmer environment whose imitator has short legs. With our implementation that uses a small neural network (due to the complexity concern mentioned above), this algorithm can obtain 39 points in 100 test episodes, while our method can obtain more than 300 points.

---

### Author Response · Authors · 2019-11-15
**revision uploaded**

We have uploaded a revision with modified part marked in red. When the whole section is newly created (Appendix D), we mark the title as red.

---

### Decision · Program_Chairs · 2019-12-19

**Decision:**

Accept (Poster)

**Comment:**

This paper seeks to adapt behavioural cloning to the case where demonstrator and learner have different dynamics (e.g. human demonstrator), by designing a state-based objective. The reviewers agreed the paper makes an important and interesting contribution, but were somewhat divided about whether the experiments were sufficiently impactful. They furthermore had additional concerns regarding the clarity of the paper and presentation of the method. Through discussion, it seems that these were sufficiently addressed that the consensus has moved towards agreeing that the paper sufficiently proves the concept to warrant publication (with one reviewer dissenting).

I recommend acceptance, with the view that the authors should put a substantial amount of work into improving the presentation of the paper based on the feedback that has emerged from the discussion before the camera ready is submitted (if accepted).